# Characterizing nitrogen cycling microorganisms and genes in sediments of the Three Gorges Reservoir

Chang Han[1], Zhiyuan Chen[1], Yihui Xiao[1], Ting Yang[2], Haoyang Shi[2], Huiqun Cao[2], Wenjun Yang[2]*, Ping Gong[2]

1 College of Water Conservancy and Hydropower Engineering, Hohai University, Nanjing, China,
2 Changjiang River Scientific Research Institute, Wuhan, China

* yangwj@mail.crsri.cn

## Abstract

Microorganisms play a central role in driving the biogeochemical cycles in lakes (reservoirs). This study aims to refine the microbial-driven nitrogen cycle processes in the sediments of the Three Gorges Reservoir and assess the overall state of nitrogen cycling within these sediments. The study focuses on the Three Gorges Reservoir as the research area, using metagenomic sequencing as a research method and measuring various environmental factors in the sediment of the region, systematically investigates the nitrogen cycle microorganisms and corresponding functional gene abundance characteristics attached to sediments from upstream, midstream, and downstream areas within the region, and explores key factors that may influence the composition of nitrogen cycle microbial communities. The outcomes of the present study manifest that within the sediments of the Three Gorges Reservoir, seven principal nitrogen cycling pathways exist. These pathways are specifically nitrogen fixation, nitrification, denitrification, nitrogen transport, organic nitrogen metabolism, assimilatory nitrate reduction, and dissimilatory nitrate reduction. Furthermore, the results of this study also reveal that the anaerobic ammonium oxidation genes are barely present in the sediments of this region, which indicates that the probability of the occurrence of anaerobic ammonium oxidation reactions in this area is negligible. The abundance of nitrogen cycle related functional genes and the diversity, composition and community structure of nitrogen cycling microorganisms differ among the upstream, midstream, and downstream regions. This suggests that as sediment particle size decreases along the course from the upstream to the downstream, it may have an impact on the distribution and community structure of nitrogen cycling microorganisms.

**Data availability statement:** The minimal dataset required to replicate the study findings has been deposited in Science Data Bank (ScienceDB), a public repository that ensures long-term preservation and accessibility. The dataset is freely available under the CC BY-NC-SA 4.0 license and can be accessed via the following persistent identifiers and link: A. Location of the data: Science Data Bank (ScienceDB) B. DOI/accession number and direct link: DOI: 10.57760/sciencedb.24483 Direct link: https://cstr.cn/31253.11.sciencedb.24483 This deposition complies with PLOS' open data policy, providing unrestricted access to the data necessary for validation, replication, and reanalysis of the study results.

**Funding:** The authors gratefully acknowledge the financial support provided by the Key Project of the National Natural Science Foundation of China (52130903). The funder had no role in study design, data collection and analysis, decision to publish, or preparation of the manuscript.

**Competing interests:** The articles I submit are completely original, without any act of plagiarism or improper quotation. The articles submitted in this article are completed by me in the process of study, research and work, and there is no conflict of interest. I do not receive any relevant honors and compensation for this. I have clearly understood that the copyright of the submitted articles belongs to this journal, and this journal has the right to copy, disseminate and digitally store the relevant articles. I have clearly understood that the publication of my articles in this journal indicates the approval of this article, but the articles published in this journal are for academic communication only and shall not be used for commercial purposes. I guarantee that the articles submitted have not been published in academic institutions at home and abroad, and will not be re-submitted or reproduced in any other academic institutions during or after the submission. I guarantee that if the submitted article has been submitted or published in other journals, I should provide the relevant publication certificate and clearly indicate the difference between the cited literature and this article, so as to prevent the relevant article from bringing relevant conflicts of interest when it is published in this journal. If there is any necessary copyright statement in

# 1 Introduction

Microorganisms involved in the nitrogen cycle play a primary role in driving the nitrogen cycle in lakes (reservoirs), influencing the generation and characteristics of various forms of nitrogen [1]. They play a significant role in the biogeochemical cycling of nitrogen within the lake (reservoir) ecosystem [2]. The sediment in lakes (reservoirs) accumulates abundant nutrients and organic matter, which can provide food for nitrogen cycle microorganisms, attract microbial adhesion, and promote their growth and metabolism [3]. Moreover, the sediment contains a strong gradient of environmental change, providing more ecological niches for nitrogen cycle microorganisms, leading to stronger functional activity of nitrogen cycle-related microorganisms [4], influencing the nitrogen cycle process in lakes (reservoirs) [5]. The high density of microorganisms attached to the sediment surface facilitates interactions among them, facilitating the acquisition of functional traits, species differentiation, and evolution of nitrogen cycle microbial species [6,7], leading to interactions among various processes of the nitrogen cycle [8]. In recent years, the unique role of sediment in the nitrogen cycle process of lakes (reservoirs) has attracted the attention of many researchers.

Deposits from various regions and depths, due to their distinct compositions, origins, and redox conditions [9], exert a filtering effect on the species composition, community structure, and functional genes of nitrogen cycling microorganisms [10], leading to the occurrence of various nitrogen transformation processes or alterations in the intensity of nitrogen reaction processes on the deposits [9]. However, current research on the nitrogen cycle processes driven by microorganisms attached to lake (reservoir) sediments often overlooks the impact of sediment particle size on the nitrogen cycle processes on particulates [8,9]. Moreover, previous studies have primarily focused on a single nitrogen cycle process [9], with limited research on the overall nitrogen cycle process, resulting in researchers lacking a clear understanding of the complete nitrogen cycle process in lake (reservoir) sediments, neglecting potential interactions between different functional groups of microorganisms. This may lead to inaccurate assessments of the nitrogen cycle function in lake (reservoir) ecosystems.

Due to the gradual decrease in flow velocity from the upstream to the downstream of the Three Gorges Reservoir, the hydrodynamic intensity correspondingly diminishes, resulting in an overall trend of decreasing sediment particle size along the reservoir from upstream to downstream. This condition has created a natural gradient of sediment particle sizes from upstream to downstream in the Three Gorges Reservoir, providing an excellent opportunity to study the distribution of nitrogen cycle functional genes and the diversity, composition, and community structure of nitrogen cycling microorganisms under different particle size conditions. Therefore, in this study, nine representative sampling sites were selected in the Three Gorges Reservoir. Metagenomic sequencing technology was utilized to investigate the distribution characteristics of nitrogen cycling functional genes, the diversity, composition and community structure of nitrogen cycling microorganisms, as well as their mutual relationships with environmental factors. We found that the distribution and composition of nitrogen cycle functional genes and nitrogen cycling microorganisms vary in the

upstream, midstream, and downstream of the reservoir. This suggests that different sediment particle sizes may have an impact on nitrogen cycling microorganisms and the nitrogen cycle processes. Therefore, this study not only reveals the nitrogen cycling mechanism in the sediments of the Three Gorges Reservoir, but also has certain significance for clarifying the nitrogen cycling mechanism of lakes (reservoirs) under different sediment particle size conditions. It can provide a theoretical basis for the prevention and control of nitrogen pollution in lakes (reservoirs) under different sediment particle size conditions.

## 2 Materials and methods

### 2.1 Overview of the study area

The study area encompasses the Three Gorges Reservoir Region (Fig 1), a 20-county area impacted by the Three Gorges Project and designated for resettlement. Data from the Hydrology Bureau of the Changjiang Water Resources Commission of the Ministry of Water Resources of the People's Republic of China, indicate that the sediment particle size in the Three Gorges Reservoir shows a decreasing trend from upstream to downstream. Specifically, from the Cuntan Hydrological Station to the Huanglingmiao Hydrological Station, the median particle size of sediments decreases from 0.01 mm to 0.006 mm. In the section from Fengdu County, Chongqing City to the Three Gorges Dam, the range of the median particle size of sediments is approximately between 0.005 mm and 0.1 mm. Among them, the median particle size of sediments in the section in front of the dam is below 0.01 mm. Consequently, nine sampling points were selected from Fengdu to the dam section and categorized as upstream, midstream, and downstream regions. The upstream sampling points include Xingyi Town, Fengdu County, Chongqing (S240), Wuyang Town, Zhong County, Chongqing (S219), and Zhongzhou Town, Zhong County, Chongqing (S206); midstream sampling points include Shuangjiang Street, Yunyang County, Chongqing (XJ001), Yongle Town, Fengjie County, Chongqing (S113), and Longmen Street, Wushan County, Chongqing (DN001); and downstream sampling points include Guizhou Town, Gui County, Yichang City, Hubei Province (XX001), Guizhou Town, Zigui County, Yichang City, Hubei Province (S39), and Maoping Town, Zigui County, Yichang City, Hubei Province (S32).

### 2.2 Sample collection method

Using a column-mounted sediment sampler, nine sampling sites were collected from 0–30 cm of sediment. Each sampling site was sampled with three tubes of sediment, and the sediment in each tube was divided into three equal portions, corresponding to depths of 0–10 cm, 10–20 cm, and 20–30 cm. The samples from the same depth in each tube were then uniformly mixed. Finally, all samples were placed in a 4°C storage box for preservation and promptly transported back to the laboratory. One portion of the samples was kept refrigerated at 4°C for determination of physicochemical properties; another portion was freeze-dried and then passed through a 100-mesh nylon sieve, and then stored frozen at −80°C for DNA extraction and molecular biology experiments.

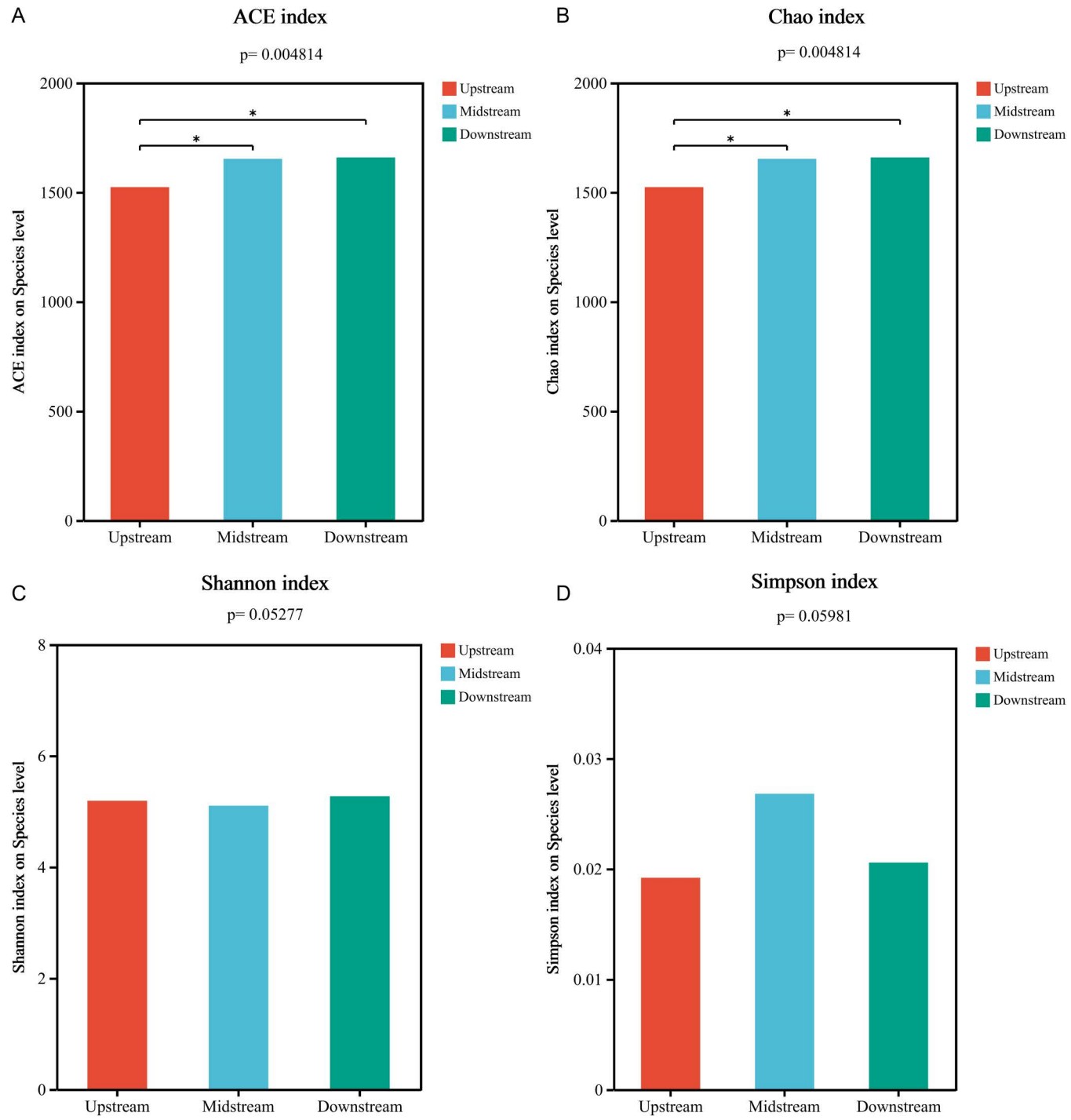

**Fig 1. Presents the microbial α diversity indices of the nitrogen cycle in sediments from the Three Gorges Reservoir.** (a) The ACE index; (b) The Chao index; (c) The Shannon index; and (d) The Simpson index.

## 2.3 Determination and analysis of physicochemical properties

The moisture content (MC) and organic matter (OM) were determined using the loss of ignition (LOI) method [11]. The pH value was measured after adding ultrapure water to the sediment at a mass ratio of 5:1 of ultrapure water to sediment and leaching for 1 hour. The sediment samples were leached with a 2 mol·L$^{-1}$ KCl solution [12], and the ammonia nitrogen (NH$_4^+$-N) and nitrate nitrogen (NO$_3^-$-N) concentrations in the sediment were determined using a continuous flow analyzer (SAN plus, Skalar Analytical B.V., Breda, the Netherlands). The total carbon (TC) and total nitrogen (TN) were measured by burning the freeze-dried samples through a 100-mesh sieve, wrapped in aluminum foil, and then placed into an elemental analyzer (Elementar Vario EL III analyzer, Germany) for combustion determination. The total organic carbon (TOC) analyzer was used to automate the determination of dissolved organic carbon (DOC). All samples were subjected to three parallel measurements and the mean values were taken.

## 2.4 Metagenomic sequencing and bioinformatics analysis

DNA was extracted from samples using the FastDNA Spin Kit for Soil (MP Biomedicals). The quality of extracted DNA was verified by 1% agarose gel electrophoresis, Quantus Fluorometer (PicoGreen) for concentration measurement, and NanoDrop 2000 for purity assessment. Qualified DNA was fragmented (Covaris M220) and subjected to Illumina paired-end sequencing at Shanghai Majorbio Bio-pharm Technology Co., Ltd. Raw sequencing data underwent quality control with fastp, followed by sequence assembly using Megahit (with iterative optimization of k-mer parameters). Open reading frames (ORFs) were predicted from the assembled contigs via Prodigal, and genes with lengths ≥ 100 bp were retained for downstream analysis. To construct a non-redundant gene set, CD-HIT was employed to cluster predicted genes from all samples (thresholds: 90% identity, 90% coverage), with the longest gene in each cluster selected as the representative sequence. High-quality reads from each sample were aligned to the non-redundant gene set using SOAPaligner (95% identity) to calculate gene abundance. Taxonomic annotation was performed by aligning the gene set to the NR database via BLASTP (e-value ≤ 1e-5), and species abundance was summarized across eight taxonomic ranks (Domain to Species). Functional annotation was conducted using KOBAS 2.0 against the KEGG database, with enrichment analysis based on gene abundances of KEGG Orthology (KO), pathways, and modules. For nitrogen cycle-related genes, sequences corresponding to KO identifiers were extracted from the non-redundant gene set, followed by taxonomic/functional annotation and abundance profiling.

## 2.5 Data analysis

In this study, we utilized R language for the creation of bar charts, pie charts, and heatmaps. We employed R language for Principal Component Analysis (PCA) and the generation of PCA diagrams. We utilized R language for Analysis of Molecular Variance (ANOSIM) and the production of box plots. We applied the Kruskal-Wallis Rank Sum Test and created multiple group comparison diagrams using R language. We utilized LEfSe software for the generation of LEfSe diagrams. Based on KEGG annotation results, we conducted differential testing and visual analysis of enzymes with differential abundance in the nitrogen metabolism pathway. We employed iPath 3 for the creation of iPath metabolic pathway diagrams. Network analysis diagrams were created using Gehpi0.10.1. Relevant diagrams were prepared using Origin2024 and Microsoft Office 2021.

## 2.6 Additional information

No permits were required for this study as the research was conducted in public areas and within regions that do not necessitate specific authorization for the type of data collection and analysis we performed. Our work adhered to all ethical guidelines and did not involve any activities that would require special permissions or approvals.

## 3  Results and analysis

### 3.1  Analysis of alpha diversity

The ACE index, based on species richness considering relative abundance, is used to estimate species richness, especially with uneven richness distribution. The Chao index, a non – parametric method, estimates unobserved species from observed species and rare species. As in Fig 1a, b, nitrogen cycling microorganism richness in the Three Gorges Reservoir increases from upstream to downstream, with significant differences between middle – upstream and upstream – downstream regions. The Shannon index combines species richness and evenness. A higher Shannon index means greater diversity and evenness in a community, and it weighs all species' relative abundance equally, being sensitive to both rare and common species. The Simpson index focuses on species abundance and dominance in a community. A smaller Simpson index indicates higher diversity, and it's more sensitive to dominant species, better reflecting uneven species distribution. As shown in Fig 1c, d, for nitrogen cycling microorganisms in the Three Gorges Reservoir, there's no significant change in evenness from upstream to downstream, and no significant differences among upstream, middle, and downstream regions.

### 3.2  Analysis of species and functional composition

Utilizing KEGG data for functional gene annotation, a total of 46 nitrogen cycle functional genes were identified, encompassing eight nitrogen cycle functions: nitrogen fixation, nitrification, denitrification, anaerobic ammonia oxidation, nitrogen transport, organic nitrogen metabolism, assimilative nitrate reduction, and dissimilatory nitrate reduction. Notably, the relative abundance of genes for anaerobic ammonia oxidation is extremely low (nearly zero), indicating a significantly low metabolic potential for this process in the sediment, and potentially even the absence of anaerobic ammonia oxidation (Fig 2a). Upon comprehensive analysis of the relative abundance of microbial nitrogen cycle genes obtained from the annotation, it was observed that in the sediment upstream, midstream, and downstream of the Three Gorges Reservoir, the highest relative abundance of nitrogen cycle genes was observed in the organic nitrogen metabolism function, accounting for 46%−53% of all nitrogen cycle genes. This was followed by denitrification genes, which comprised 25%−26% of all functional genes in each group. The next most abundant categories were nitrogen transport genes (9%−14%), followed by dissimilatory nitrate reduction genes (6%−8%), assimilative nitrate reduction genes (2%−4%), nitrification genes (3%−3%), and finally nitrogen fixation genes (1%−2%). From Fig 2a, it can also be observed that from the upstream to the midstream, the relative abundance of functional genes involved in organic nitrogen metabolism decreases, while the relative abundance of functional genes involved in nitrogen transport increases. In addition to these two types of functional genes showing significant changes in relative abundance, the relative abundance of several other functional genes related to nitrogen cycling changes minimal from the upstream to the midstream. From the midstream to the downstream, there are only minor changes in the relative abundance of functional genes involved in organic nitrogen metabolism, nitrogen transport, assimilatory nitrate reduction, dissimilatory nitrate reduction, and nitrogen fixation, while there is no change in the relative abundance of functional genes involved in denitrification and nitrification. As depicted in Fig 2b–d, when classifying microorganisms by their taxonomic groups, the highest relative abundance of microorganisms in the upper, middle, and lower reaches of the Three Gorges Reservoir is observed in the phylum Proteobacteria. The relative abundance of Proteobacteria is significantly higher than that of other microbial groups, thereby occupying an absolute dominant position.

### 3.3  Comparative analysis of species and function

As depicted in Fig 3a, an upward trend in the beta diversity of nitrogen cycling microorganisms is evident in the sediment from the upstream to the downstream regions. Fig 3b illustrates that upstream samples display a higher degree of aggregation. In contrast, midstream and downstream samples exhibit greater dispersion and are not clearly distinguished from each other.

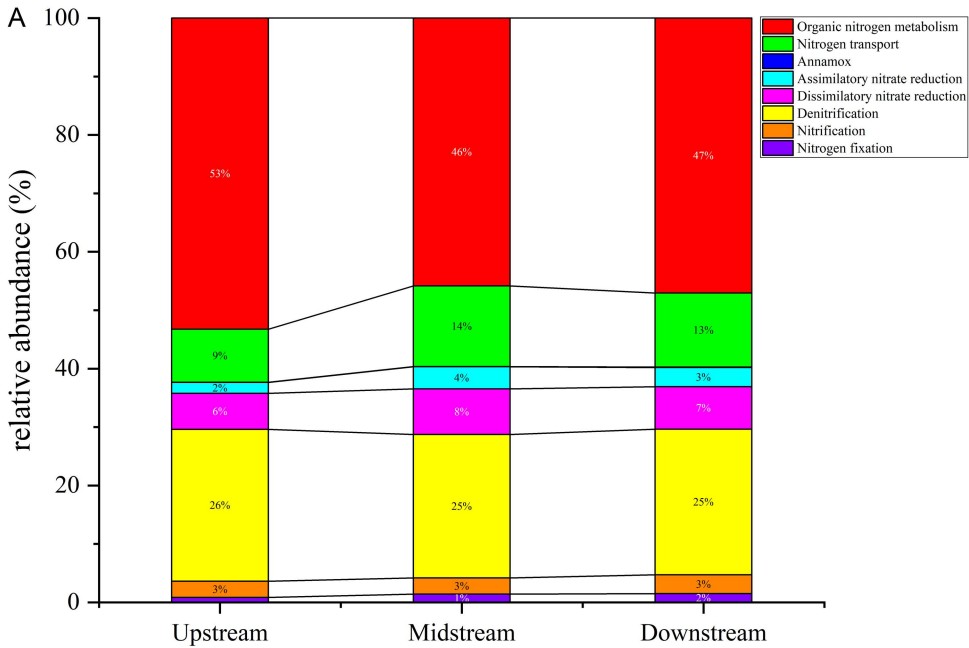

Community analysis pieplot on Phylum level：Upstream

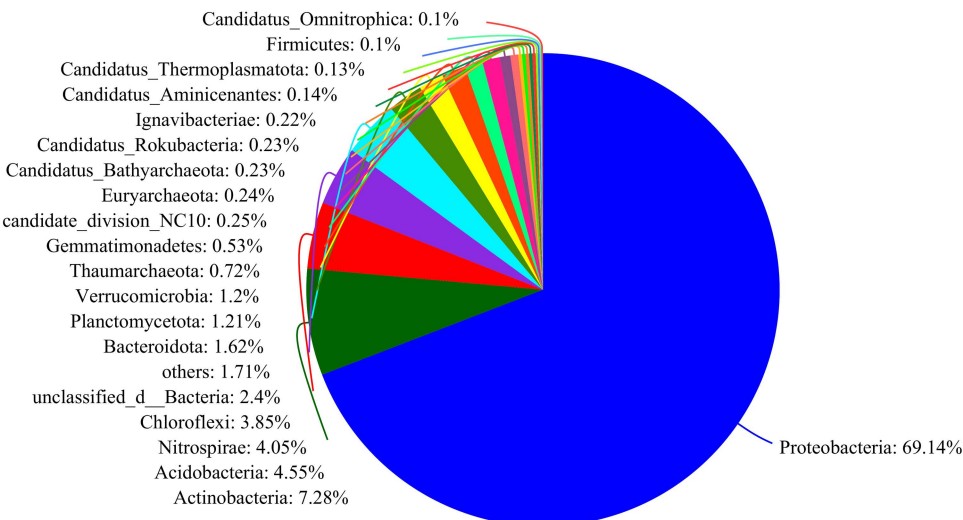

**Fig 2. Presents an analysis of the microbial species and functional composition of the nitrogen cycle in the sediments of the Three Gorges Reservoir.** (a) Presents a bar graph based on the functional gene level; (b) Presents a pie chart based on the phylum level (upstream); (c) Presents a pie chart based on the phylum level (midstream); and (d) Presents a pie chart based on the phylum level (downstream).

### 3.4 Analysis of species and functional differences

The analysis of microbial communities using LEfSe reveals distinct species at the phylum to class level, which are visually represented in the upstream, midstream, and downstream regions. As can be observed from Fig 4a, only Fibrobacteres shows no significant differences among the three groups from upstream to downstream. The comparison of

C Community analysis pieplot on Phylum level：Midstream

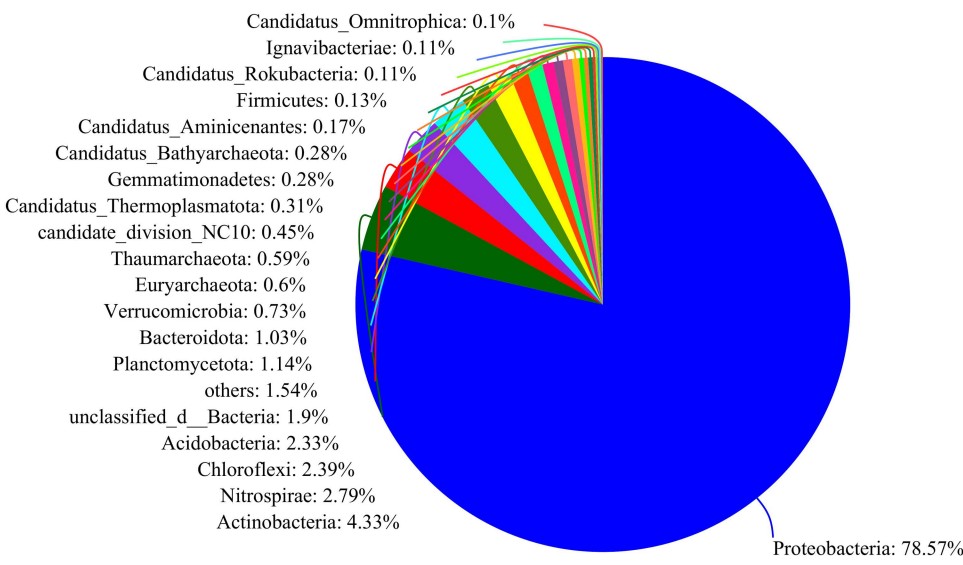

D Community analysis pieplot on Phylum level：Downstream

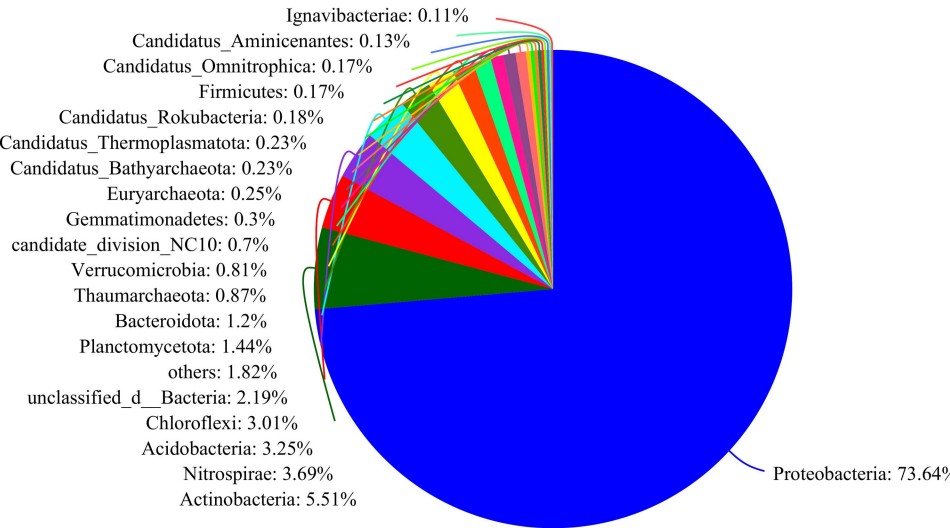

**Fig 2.** Continued.

multiple groups is primarily used to analyze whether there are differences in microbial composition among the groups and to identify microorganisms with significant differences. Fig 4b shows the differences in average relative abundance of the same species across different groups, with differences being labeled. At the species level, species with significant differences between upstream, midstream, and downstream include Moraxellaceae_bacterium, Betaproteobacteria_bacterium, Burkholderiales_bacterium, Aquabacterium_sp., Gammaproteobacteria_bacterium, Burkholderiaceae_bacterium,

A

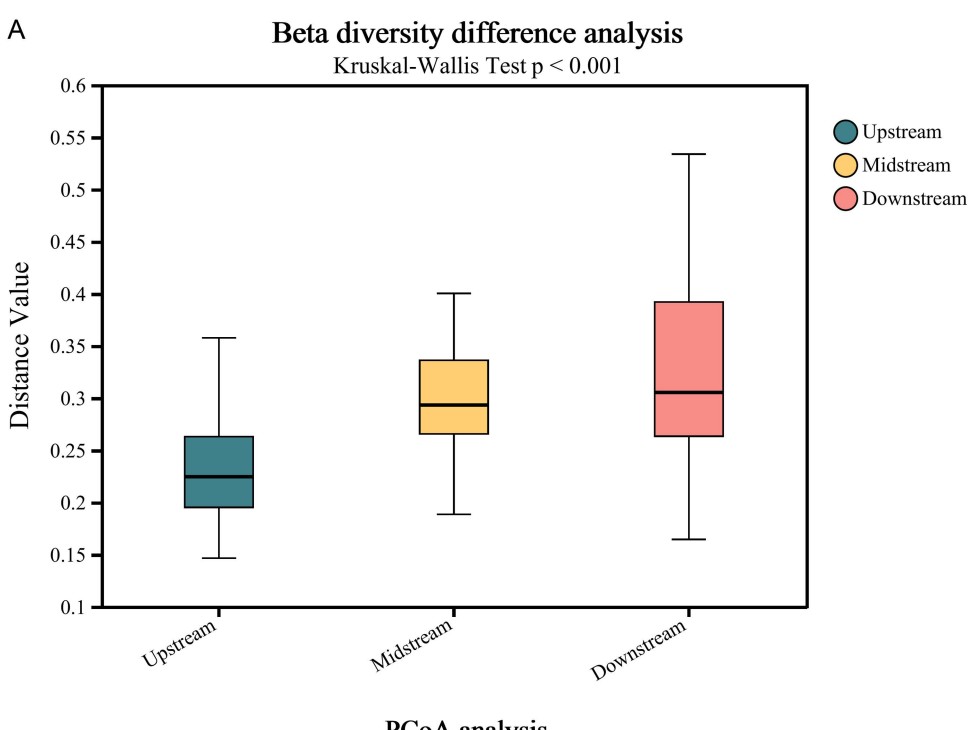

B

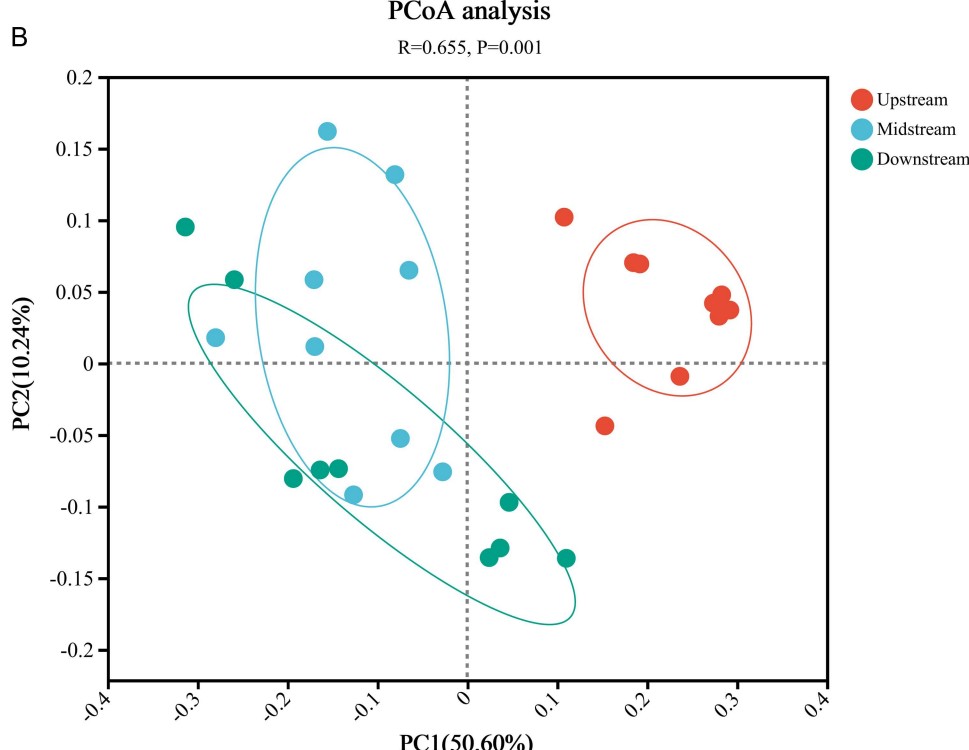

**Fig 3. Beta diversity analysis.** (a) Analysis of inter group differences in β diversity; (b) Principal Co-ordinates Analysis.

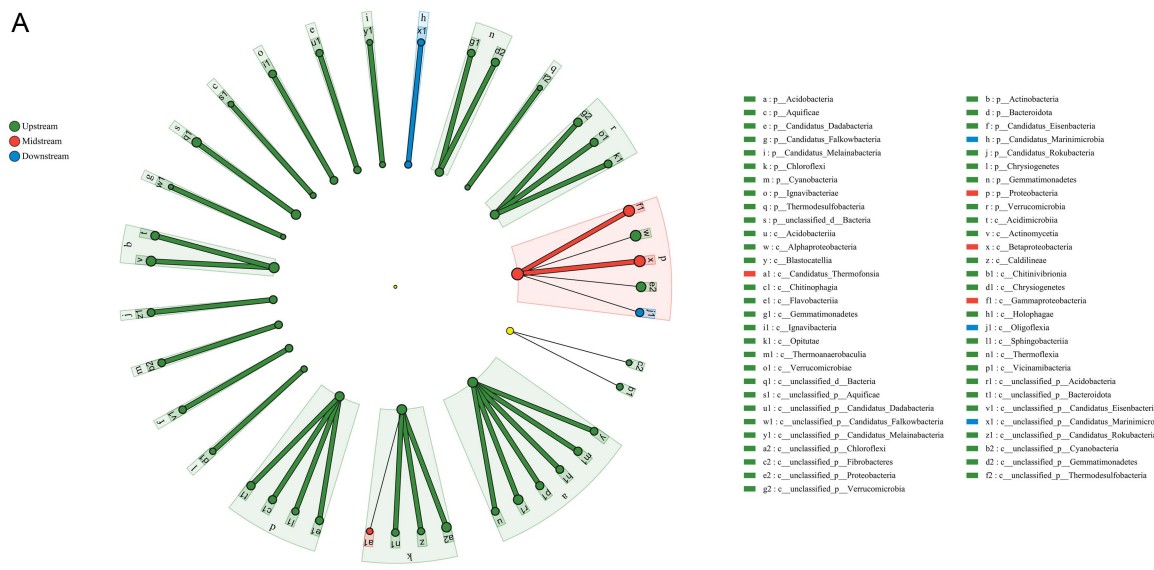

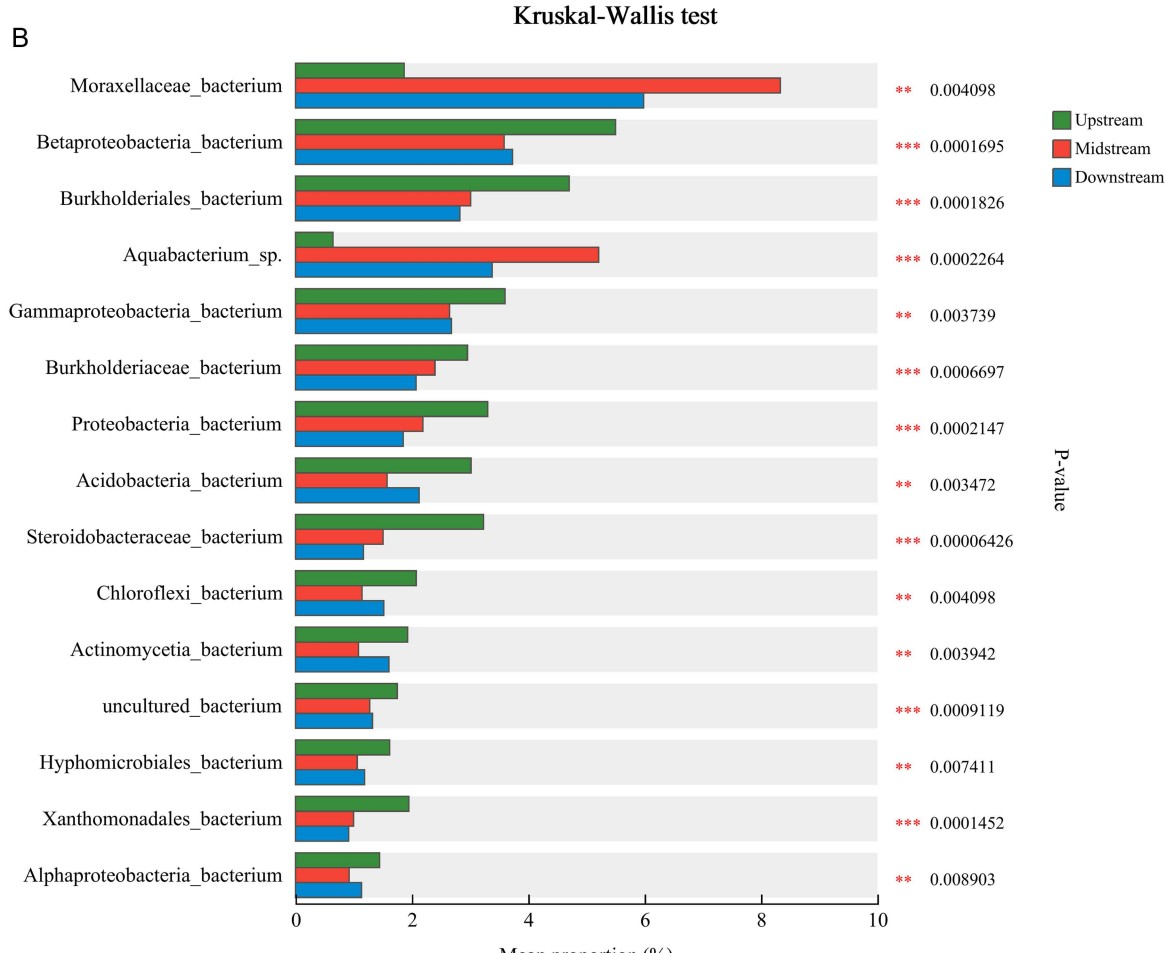

**Fig 4. Presents an analysis of the species and functional differences in nitrogen cycling microorganisms within the sediments of the Three Gorges Reservoir.** (a) Presents a LEfSe differential discrimination analysis conducted at the level of phyla to classes; (b) Presents a comparative analysis of multiple groups at the species level.

Proteobacteria_bacterium, Acidobacteria_bacterium, Steroidobacteraceae_bacterium, Chloroflexi_bacterium, Actinomycetia_bacterium, uncultured_bacterium, Hyphomicrobiales_bacterium, Xanthomonadales_bacterium, and Alphaproteobacteria_bacterium.

### 3.5 Correlative analysis of environmental factors

The Mantel Test network heatmap analysis is carried out by performing a correlation analysis between two matrices. In the field of microbiomics, it is frequently employed to evaluate the correlation between environmental factors and the structure of microbial communities, and the species are categorized at the taxonomic level. As shown in Fig 5, the lines in the figure depict the correlation between communities and environmental factors, whereas the heatmap presents the correlation among environmental factors. It can be clearly seen from the figure that the upstream area exhibits a positive correlation with MC, OM, $NO_3^-$, and TP, and a notably significant positive correlation is detected with MC. Moreover, it shows a negative correlation with pH, $NH_4^+$, TN, and DOC. Regarding the midstream area, it has a positive correlation with pH, MC, OM, $NH_4^+$, $NO_3^-$, TN, TP, and DOC. As for the downstream area, it is positively correlated with MC, OM, $NH_4^+$, TN, and DOC, but shows a negative correlation with pH, $NO_3^-$, and TP.

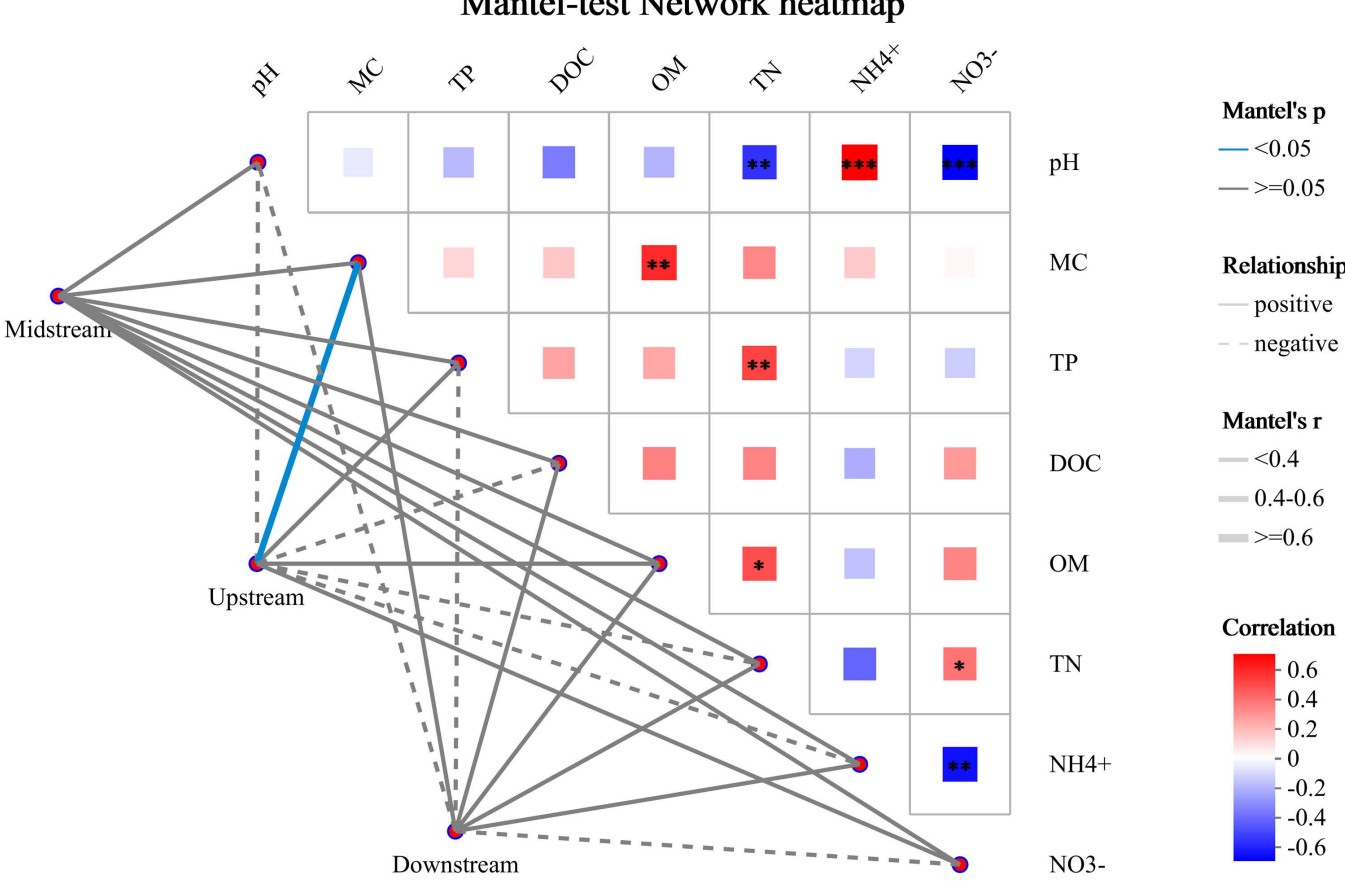

**Fig 5. Mantel-test network heatmap at the species level.**

### 3.6 Random forest analysis

The diagram (Fig 6) based on the importance rankings of random forest-derived flagship species and environmental factors reflects the influence of different species and environmental factors on the accuracy of the classification model. In this diagram, the x-axis (Mean Decrease Accuracy) functions as a measure of importance, where higher values denote greater significance. The y-axis presents the names of species and environmental factors arranged in order of importance. At the species level, the flagship species comprise: Tepidicella_xavieri, Azonexus_sp., Leptothrix_sp._(in:_Bacteria),

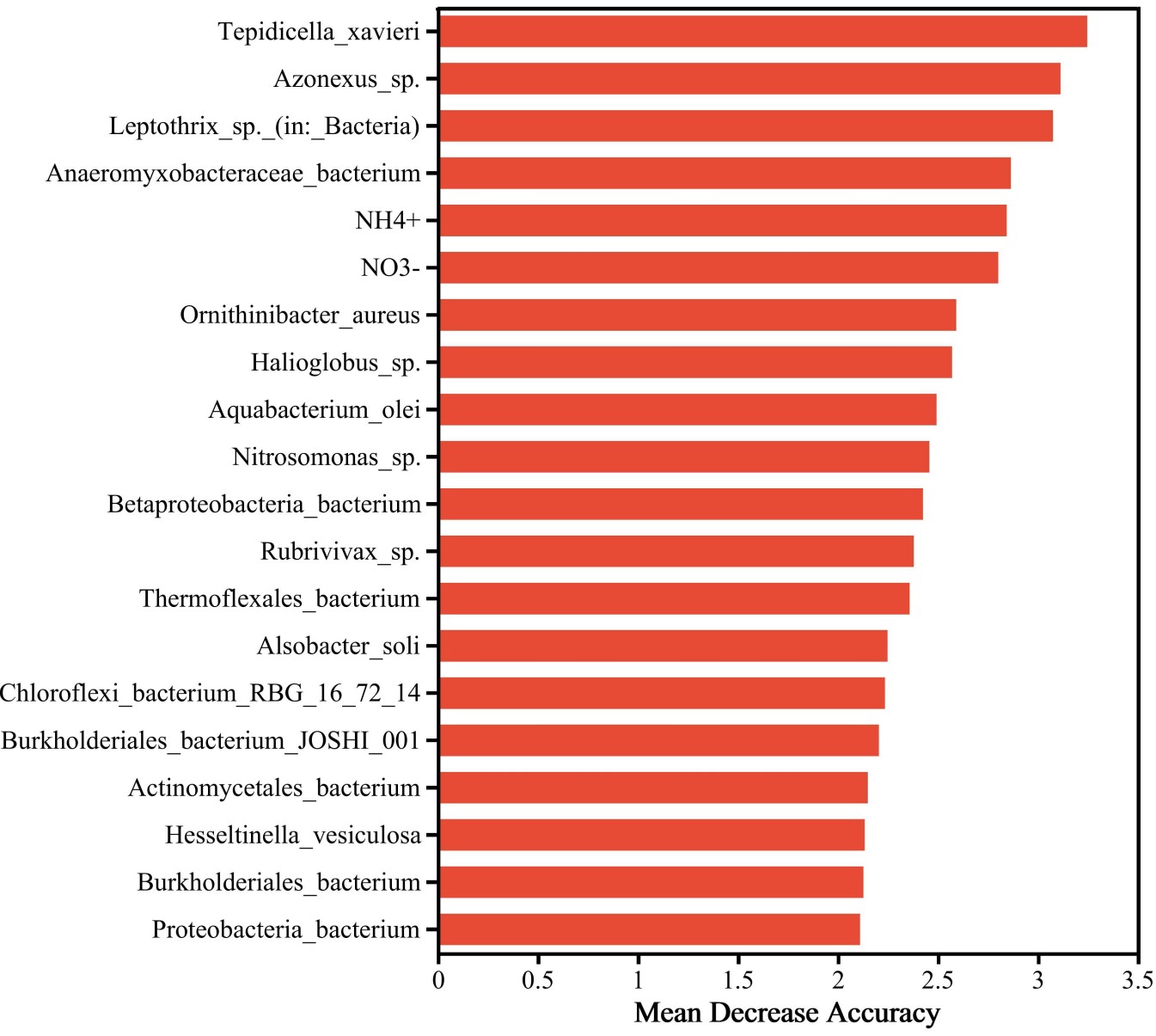

**Fig 6. Analysis of random forests.**

Anaeromyxobacteraceae_bacterium, Ornithinibacter_aureus, Halioglobus_sp., Aquabacterium_olei, Nitrosomonas_sp., Betaproteobacteria_bacterium, Rubrivivax_sp., Thermoflexales_bacterium, Alsobacter_soli, Chloroflexi_bacterium_ RBG_16_72_14, Burkholderiales_bacterium_JOSHI_001, Actinomycetales_bacterium, Hesseltinella_vesiculosa, Burk-holderiales_bacterium, Proteobacteria_bacterium. The flagship environmental factors are $NH_4^+$ and $NO_3^-$.

### 3.7 Network analysis

Based on the nitrogen cycle functional genes, a network analysis diagram of samples from the upstream, midstream, and downstream of the Three Gorges Reservoir is constructed. As seen in Fig 7a, the eight nitrogen transformation

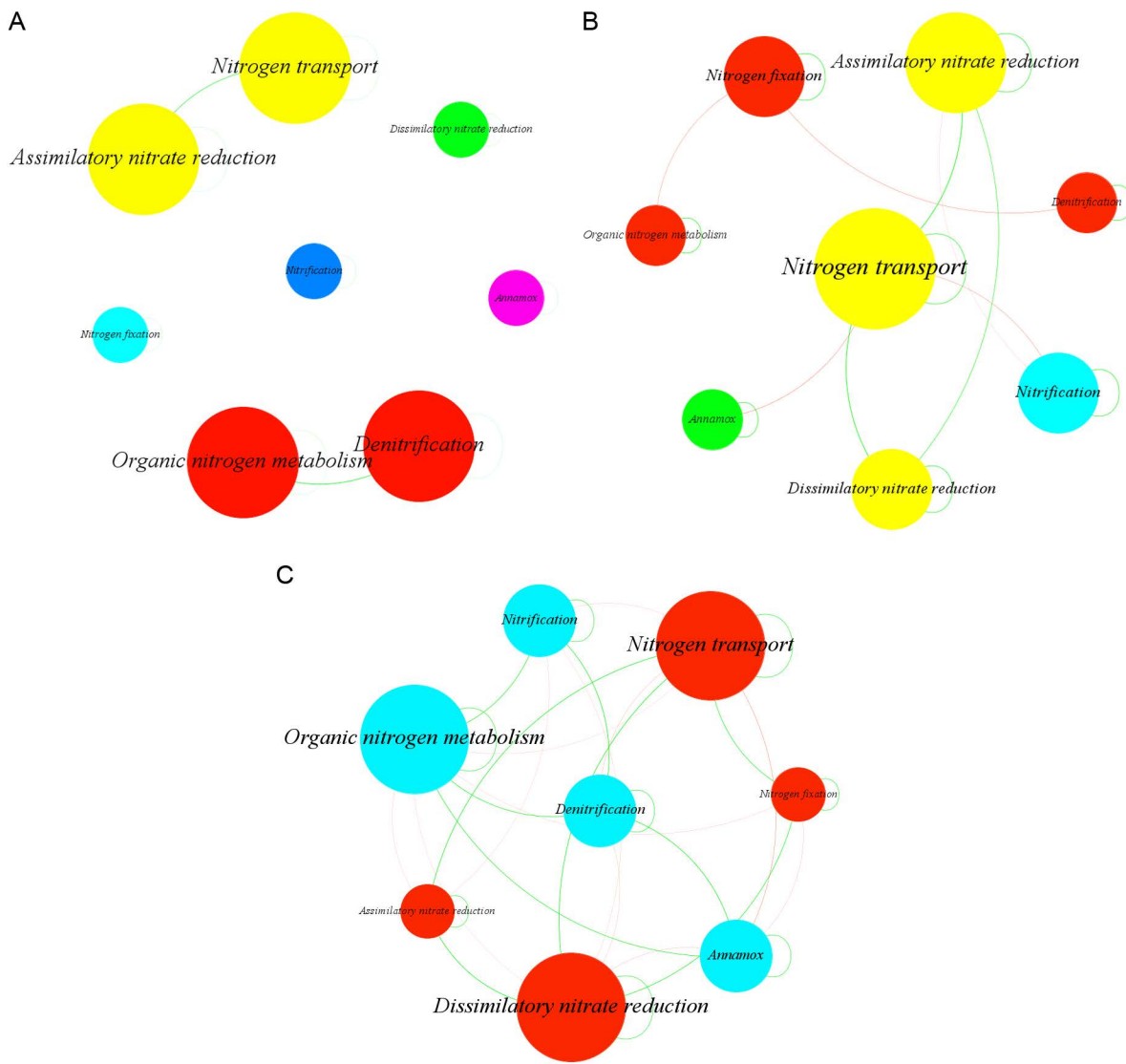

**Fig 7. Presents the network analysis diagrams.** (a) is based on the functional gene level, referred to as the upstream region; (b) is based on the functional gene level, referred to as the midstream region; (c) is based on the functional gene level, referred to as the downstream region; (d) is based on the phylum level, referred to as the upstream region; (e) is based on the phylum level, referred to as the midstream region; (f) is based on the phylum level, referred to as the downstream region.

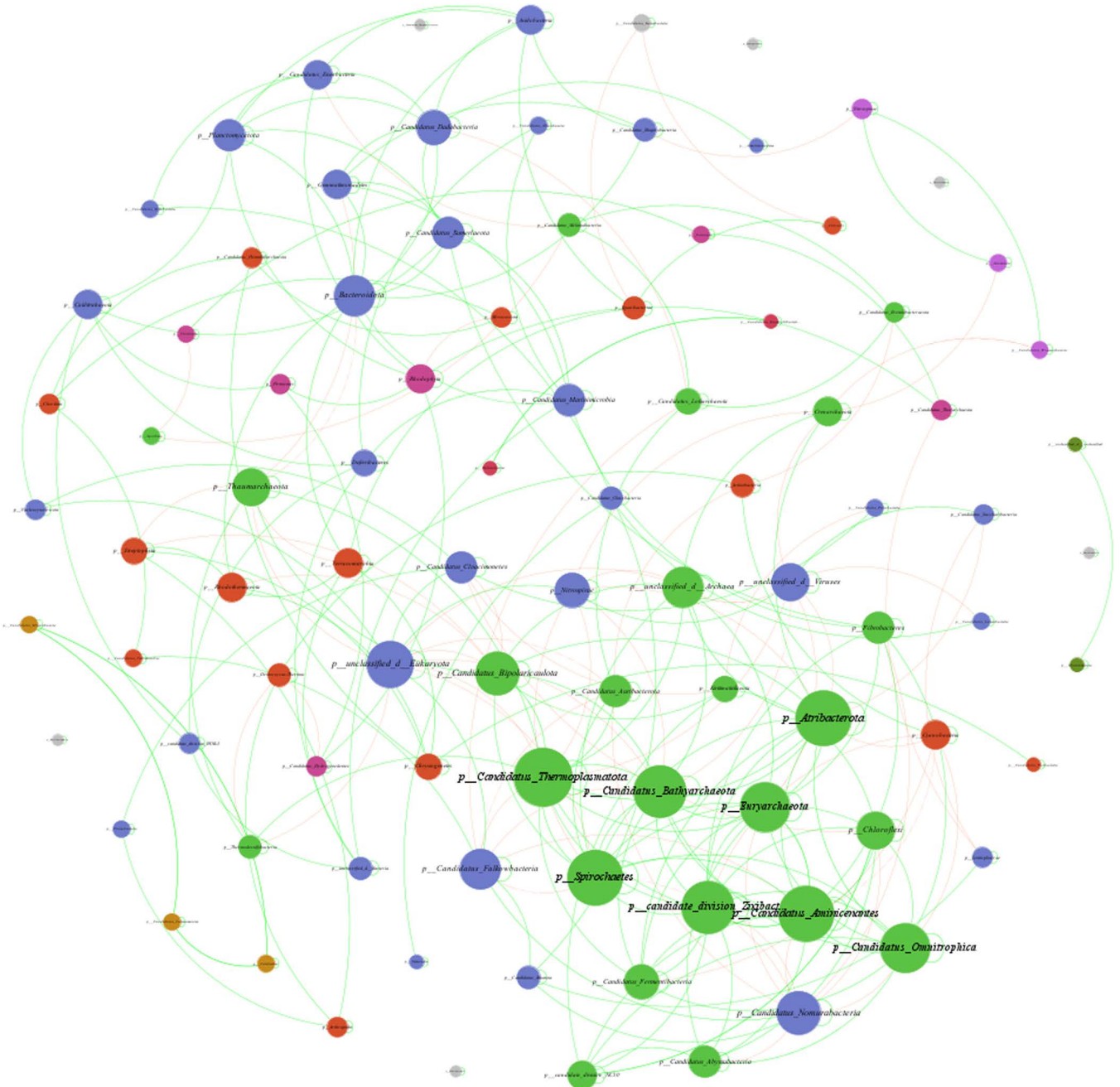

**Fig 7.** Continued.

functions in upstream samples are grouped into six modules. The nitrate reduction assimilation and nitrogen transport functions form one module, making up 25% of the six. Organic nitrogen metabolism and denitrification functions cluster into one module, also 25%. Nitrogen fixation, nitrification, anaerobic ammonia oxidation, and dissimilatory nitrate reduction functions each form an independent module, with each accounting for 12.5%. All lines in the diagram show positive correlations, meaning all eight functions are positively related. From Fig 7b, the eight functions in midstream samples are grouped into four modules. Nitrate reduction assimilation, nitrogen transport, and dissimilatory nitrate reduction functions

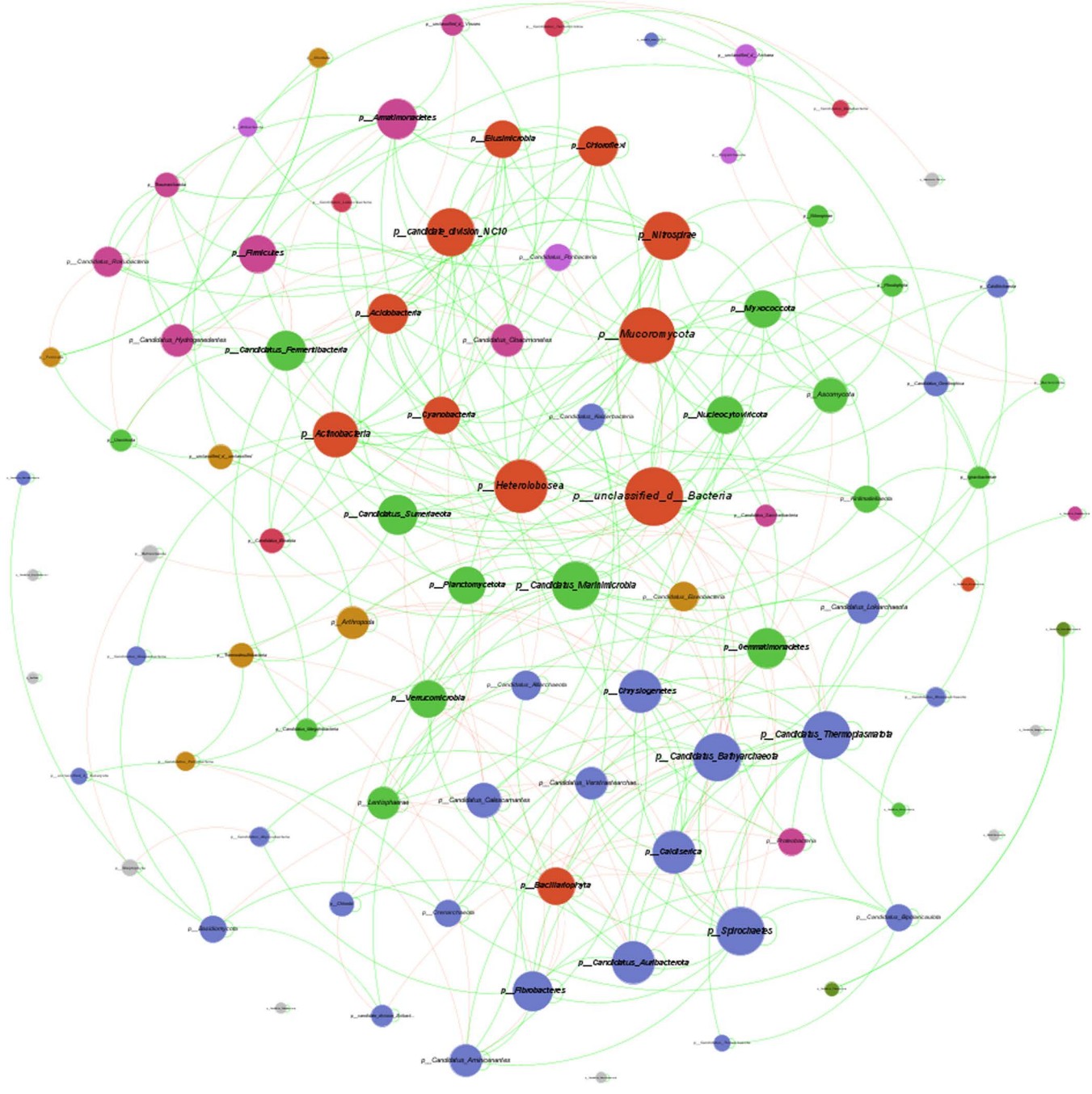

**Fig 7.** Continued.

form one module, accounting for 37.5% of the four. Nitrogen fixation, organic nitrogen metabolism, and denitrification functions cluster into one module, also 37.5%. Nitrification and anaerobic ammonia oxidation functions each form an independent module, with each accounting for 12.5%. Positive correlations make up 68.75% of the lines and negative correlations 31.25%, so positive correlations are dominant. As shown in Fig 7c, the eight functions are grouped into two modules. Nitrification, organic nitrogen metabolism, denitrification, and anaerobic ammonia oxidation functions form one module, taking

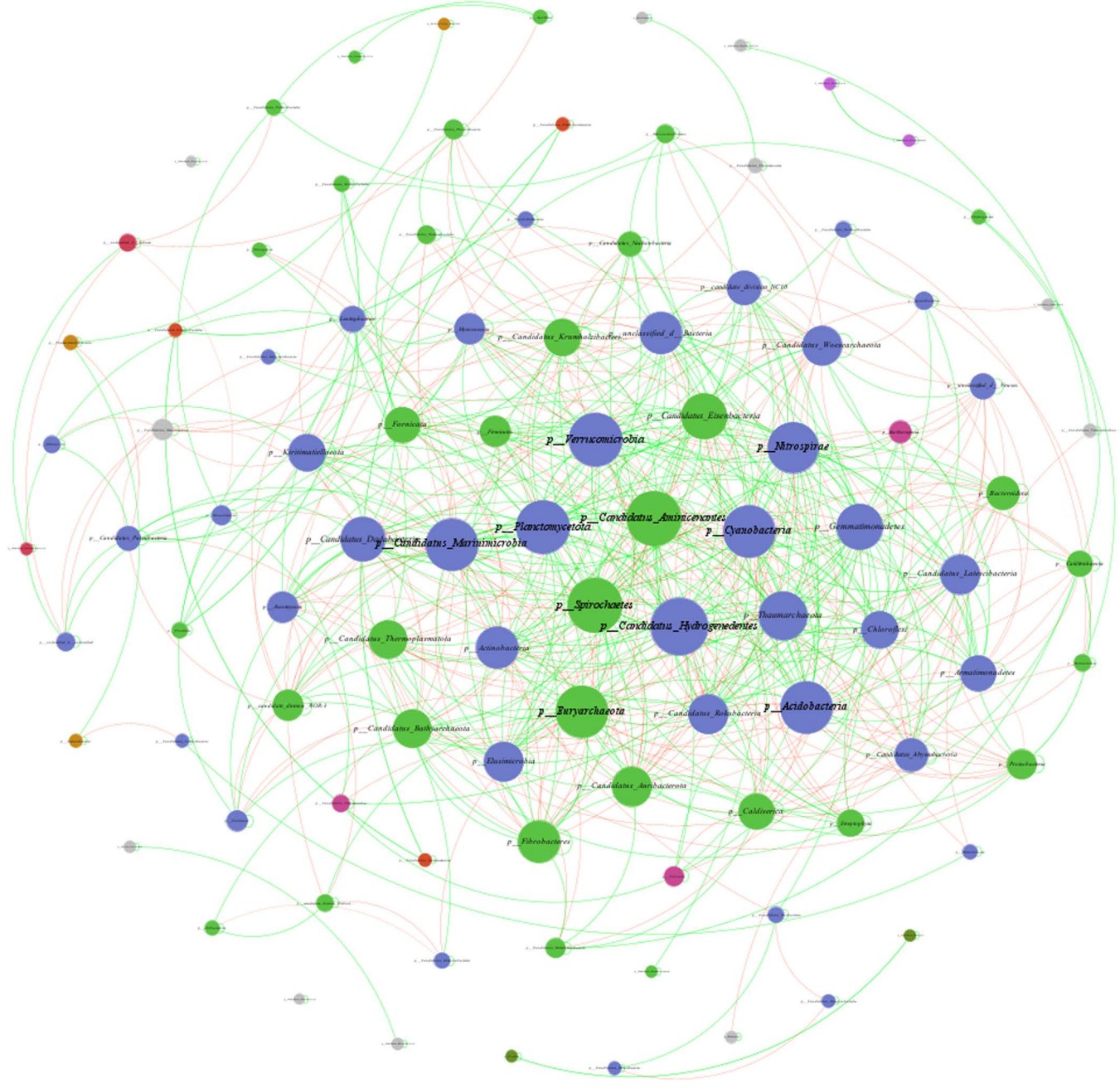

**Fig 7.** Continued.

up 50% of the two. Nitrogen transport, nitrogen fixation, nitrate reduction assimilation, and dissimilatory nitrate reduction functions cluster into one module, also 50%. Positive correlations account for 60% of the lines and negative correlations 40%, indicating positive correlations still dominate.

Based on species level analysis of samples from the upstream, midstream, and downstream of the Three Gorges Reservoir, network analysis diagrams were constructed. As seen in Fig 7d, nitrogen cycling microorganisms in upstream samples were clustered into 15 modules by species. The largest proportion module was 33.33% of all. The second largest was 25.81%, the third 16.13%, the fourth 6.45%, and the fifth and sixth each 3.23%. Modules from the seventh to the

fifteenth together accounted for 1.08%. Positive correlated links were 79.22% and negative correlated ones were 20.78%, showing positive correlations dominate among nitrogen cycling microorganisms. From Fig 7e, in midstream samples, nitrogen cycling microorganisms were grouped into 17 modules. The highest proportion module was 29.03% of all. The second highest was 19.35%, the third 12.9%, the fourth 10.75%, the fifth 7.53%, the sixth and seventh each 4.3%, the eighth 2.15%, and modules from the ninth to the seventeenth each 1.08%. Positive correlated links were 81.12% and negative correlated ones were 18.88%, indicating positive correlations are dominant. As shown in Fig 7f, for downstream samples, nitrogen cycling microorganisms were clustered into 16 modules. The module with the highest proportion was 40.4% of all. Another module was 33.33%, the third to fifth each 3.03%, and the sixth to eleventh each 2.02%. Modules from the twelfth to the sixteenth each accounted for 1.01%. Positive correlated connections were 60.46% and negative correlated ones were 39.54%, suggesting positive correlations are mainly seen among nitrogen cycling microorganisms.

## 4 Discussion

The metagenomic sequencing results indicated that seven nitrogen cycling functional genes, namely nitrogen fixation, nitrification, denitrification, nitrogen transport, organic nitrogen metabolism, assimilatory nitrate reduction, and dissimilatory nitrate reduction, were detected in the sediments of the Three Gorges Reservoir. The relative abundance of the anaerobic ammonia oxidation gene was nearly zero, indicating a minimal metabolic potential of anaerobic ammonia oxidation. Since the greenhouse gas emissions from this process are relatively low, its absence may lead to an increase in other denitrification processes such as denitrification [13]. This study results showed that the relative abundance of denitrification functional genes in the sediments of the reservoir from upstream, midstream to downstream exceeded 25%, suggesting that denitrification is the main nitrogen removal process [14]. Given that greenhouse gas $N_2O$ can be produced during denitrification [15] and that its global warming potential is much higher than that of $CO_2$, it is hypothesized that the Three Gorges Reservoir might be a significant source of greenhouse gas emissions. Future experiments will carry out a quantitative study on the $N_2O$ release rate to determine the specific emission situation in the reservoir.

A multitude of antecedent research endeavors have unequivocally demonstrated that, in contradistinction to smaller particle dimensions, larger particle sizes manifest enhanced organic nitrogen metabolic activities [16–20]. The findings of the current study unambiguously disclose that, during the transition from the upstream to the midstream of the reservoir, concomitant with the successive diminution in sediment particle size, the relative abundance of functional genes affiliated with organic nitrogen metabolism experiences a downtrend. This strongly intimates that larger sediment particles may, in fact, harbor a more substantial potential for organic nitrogen metabolism. This phenomenon could plausibly be ascribed to the propensity of organic nitrogen to adhere more readily to larger sediment particles, thereby endowing an abundant organic nitrogen substrate for microorganisms and, in turn, culminating in a relatively augmented abundance of functional genes pertinent to organic nitrogen metabolism on larger sediment particles. This study has revealed that, at the phylum level, Proteobacteria served as the predominant microorganisms responsible for driving the nitrogen cycle in the upstream, midstream, and downstream regions. This could potentially be attributed to a combination of factors, such as the extensive niche adaptability of Proteobacteria, its crucial capabilities in engaging in the nitrogen cycle processes, its interaction with organic matter, and the multiple impacts of environmental factors [21–25]. The circumstance where Proteobacteria were the dominant nitrogen cycling microorganisms across the upstream, midstream, and downstream areas implies that the alteration in sediment particle size appears to have a rather insignificant influence on the composition of nitrogen cycling microorganisms.

From the upstream to the downstream of the Three Gorges Reservoir, the diversity of nitrogen cycling microorganisms tends to increase as the sediment particle size decreases. This trend may be due to the fact that smaller sediment particles provide a larger specific surface area, offering more attachment sites for microorganisms, thereby supporting a more diverse microbial community [26–28]. Moreover, fine sediment particles can better retain moisture and nutrients, providing a rich source of nutrients for microorganisms [29,30]. Additionally, small particles can create more microenvironments, providing suitable habitats for different types of microorganisms [31–34]. From the upstream to the downstream

of the Three Gorges Reservoir, the degree of dispersion among samples within the nitrogen cycling microbiome tends to increase as the sediment particle size decreases. This trend may be due to the fact that finer sediment particle sizes lead to higher environmental heterogeneity, which can create more microspaces and varying redox conditions, thereby supporting microbial communities with different ecological niches [35–40].

## 5 Conclusion

This study, focused on the Three Gorges Reservoir as the research area, employed metagenomic sequencing technology to systematically identify the distribution characteristics of nitrogen cycle related functional genes, as well as the diversity, composition, and community structure of nitrogen cycling microorganisms in sediments from the upstream, midstream, and downstream of the reservoir. The main conclusions are as follows: (1) A total of seven major nitrogen cycle pathways were covered in the sediment samples from the Three Gorges Reservoir, which include nitrogen fixation, nitrification, denitrification, nitrogen transport, organic nitrogen metabolism, assimilatory nitrate reduction, and dissimilatory nitrate reduction. The abundance of the anaerobic ammonia oxidation functional gene was extremely low (almost zero), indicating that the potential for the anaerobic ammonia oxidation reaction to occur in this area was very low. (2) There were certain differences in the relative abundance of nitrogen cycling functional genes, the diversity, composition and community structure of nitrogen cycling microorganisms among the sediments with different particle sizes in the upstream, midstream and downstream areas, suggesting that the size of sediment particles might affect the community composition of nitrogen cycling microorganisms attached to them. (3) As the sediment particle size decreases, the α and β diversity of nitrogen cycling microorganisms increase, indicating that smaller sediment particle sizes may bring greater diversity for the nitrogen cycling microorganisms attached to them.

## Supporting information

**S1 File. Dataset.**
(ZIP)

## Author contributions

**Conceptualization:** Chang Han, Ping Gong.

**Data curation:** Chang Han.

**Formal analysis:** Chang Han, Ping Gong.

**Funding acquisition:** Wenjun Yang.

**Investigation:** Chang Han, Zhiyuan Chen, Yihui Xiao, Ting Yang, Haoyang Shi, Huiqun Cao.

**Methodology:** Chang Han.

**Project administration:** Chang Han, Ping Gong.

**Resources:** Chang Han.

**Software:** Chang Han.

**Supervision:** Chang Han, Ping Gong.

**Validation:** Chang Han.

**Visualization:** Chang Han.

**Writing – original draft:** Chang Han.

**Writing – review & editing:** Chang Han, Ping Gong.

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
