## [Decision Letter · Decision Letter 0]

Dear Dr. Yang,

Thank you for submitting your manuscript to PLOS ONE. After careful consideration, we feel that it has merit but does not fully meet PLOS ONE’s publication criteria as it currently stands. Therefore, we invite you to submit a revised version of the manuscript that addresses the points raised during the review process.

We look forward to receiving your revised manuscript.

Kind regards,

Farhan Hafeez, Ph.D.

Academic Editor

PLOS ONE

“The authors gratefully acknowledge the financial support provided by the Key Project of the National Natural Science Foundation of China (52130903).”

6. We note that Figure 1 in your submission contain [map/satellite] images which may be copyrighted. All PLOS content is published under the Creative Commons Attribution License (CC BY 4.0), which means that the manuscript, images, and Supporting Information files will be freely available online, and any third party is permitted to access, download, copy, distribute, and use these materials in any way, even commercially, with proper attribution. For these reasons, we cannot publish previously copyrighted maps or satellite images created using proprietary data, such as Google software (Google Maps, Street View, and Earth). For more information, see our copyright guidelines: http://journals.plos.org/plosone/s/licenses-and-copyright.

7. Please remove your figures from within your manuscript file, leaving only the individual TIFF/EPS image files, uploaded separately. These will be automatically included in the reviewers’ PDF.

Additional Editor Comments:

Dear Author(s),

I write to you regarding the manuscript PONE-D-24-37897 entitled "Characterizing Nitrogen-Cycling Microorganisms and Genes in Sediments of the Three Gorges Reservoir" which was submitted to PLOS One.

The reviewers have now commented on the manuscript. You will see that they are advising substantial revision of the manuscript. In addition to the reviewers’ comments, the overall write-up needs considerable improvement. When revising your work, please submit a list of changes or a rebuttal against each point being raised through track changes mode or by using bold or colored text.

Please revise the manuscript strictly according to the appended comments from the reviewers. While revising, need is to double check that all the references cited within the text have corresponding references.

Sincerely,

Reviewers' comments:

Reviewer's Responses to Questions

**Comments to the Author**

1. Is the manuscript technically sound, and do the data support the conclusions?

Reviewer #1: Yes

Reviewer #2: Yes

2. Has the statistical analysis been performed appropriately and rigorously?

Reviewer #1: Yes

Reviewer #2: Yes

3. Have the authors made all data underlying the findings in their manuscript fully available?

Reviewer #1: Yes

Reviewer #2: Yes

4. Is the manuscript presented in an intelligible fashion and written in standard English?

Reviewer #1: Yes

Reviewer #2: No

Reviewer #1: The manuscript ‘Characterizing Nitrogen-Cycling Microorganisms and Genes in Sediments of the Three Gorges Reservoir’ focuses an important topic of investigating microbial-driven nitrogen cycle processes in the sediments of the Three Gorges Reservoir and assess the overall state of nitrogen cycling. The current manuscript comes in the domain of the journal and I appreciate the authors for their research work. However, due to various issues across different sections of manuscript, it needs revision. Overall, sentence structure needs to be improved. Some of general comments/questions are listed below while more specific questions and comments are inserted in the attached PDF file.

In abstract elaborate briefly, what authors want to say when write ‘46 types of nitrogen conversion functional genes’, nitrogen conversion not accurate there. Further. it’s not ‘refine’ rather this study aims to investigate or report. Please rephrase. The current structure of the introduction is not well organized. Additionally, the last part of the introduction needs to be revised, considering the main theme/objectives and findings of the study. Materials and Methods are very descriptive, it’s more like a thesis document especially in methods section, there are a large number of sub-headings in methods. And the representation of data also needs to be extensively improved. Further, the figures could be merged, or only key data may be included instead of having huge numbers which could easily be merged. The results are explained while the Discussion section could be improved, for example in addition to discussing the results in line with previous reports, key findings need to be discussed and highlighted. The section 4 overall, and specifically with focus on ‘ammonia oxidation requires specific environmental conditions’ is also like a thesis document, much detail and redundant rather than just discussing the key findings. Another very crucial aspect of the work i.e. how the ‘variations in sediment particle size’ influences the N cycling is described but lacks in depth analyses, for example, the relative abundance of these functional guilds across various environments could also be compared to enlighten the current results. Again, like other sections of the manuscript, Discussion, and Conclusions sections also needs to be comprehensive (instead of detailed paras and redundancy at places), the focus could only be on the key findings of the work.

Reviewer #2: The research work described the Characterization of Nitrogen-Cycling Microorganisms and Genes in Sediments. The article reports interesting results, however write up of the article could be a lot better than this draft. Language of introduction could be improved. There are so many sub-sections in the methodology section which do not follow the article style. A huge number of figures are also not desired. The quality of Figure 2a is not readable. Nitrogen is a very crucial element across sediments, discussion section needs to be extensively improved considering the nitrogen importance across sediments.

**Do you want your identity to be public for this peer review?** For information about this choice, including consent withdrawal, please see our Privacy Policy

Reviewer #1: No

Reviewer #2: No

---

## [Author Response · Author response to Decision Letter 1]

23 Dec 2024

Reviewer #1: The manuscript ‘Characterizing Nitrogen-Cycling Microorganisms and Genes in Sediments of the Three Gorges Reservoir’ focuses an important topic of investigating microbial-driven nitrogen cycle processes in the sediments of the Three Gorges Reservoir and assess the overall state of nitrogen cycling. The current manuscript comes in the domain of the journal and I appreciate the authors for their research work. However, due to various issues across different sections of manuscript, it needs revision. Overall, sentence structure needs to be improved. Some of general comments/questions are listed below while more specific questions and comments are inserted in the attached PDF file.

In abstract elaborate briefly, what authors want to say when write ‘46 types of nitrogen conversion functional genes’, nitrogen conversion not accurate there. Further. it’s not ‘refine’ rather this study aims to investigate or report. Please rephrase. The current structure of the introduction is not well organized. Additionally, the last part of the introduction needs to be revised, considering the main theme/objectives and findings of the study. Materials and Methods are very descriptive, it’s more like a thesis document especially in methods section, there are a large number of sub-headings in methods. And the representation of data also needs to be extensively improved. Further, the figures could be merged, or only key data may be included instead of having huge numbers which could easily be merged. The results are explained while the Discussion section could be improved, for example in addition to discussing the results in line with previous reports, key findings need to be discussed and highlighted. The section 4 overall, and specifically with focus on ‘ammonia oxidation requires specific environmental conditions’ is also like a thesis document, much detail and redundant rather than just discussing the key findings. Another very crucial aspect of the work i.e. how the ‘variations in sediment particle size’ influences the N cycling is described but lacks in depth analyses, for example, the relative abundance of these functional guilds across various environments could also be compared to enlighten the current results. Again, like other sections of the manuscript, Discussion, and Conclusions sections also needs to be comprehensive (instead of detailed paras and redundancy at places), the focus could only be on the key findings of the work.

Reply:

I have already made modifications to all the problematic areas listed in the PDF file according to the comments.

I have rewritten the abstract section, reorganizing the language. The sentence "46 types of nitrogen conversion functional genes" has been removed, and the structure of the sentences has been reorganized to ensure accurate expression.

I have revised the last part of the introduction, incorporating the research theme/objectives and the investigation results.

I have simplified the descriptions in the "Materials and Methods" section and consolidated some overly detailed sub-headings to reduce the complexity of the methods part and ensure that each section is directly relevant and contributes to the research findings.

I have eliminated certain analyses that bear relatively low relevance to the overall conclusion and opted to present solely the key data. By doing so, I aim to ensure the effective conveyance of the information's essence while circumventing excessive and redundant data. I have endeavored to render the data presentation more concise and influential, thus conforming to the standards expected of high-quality research papers.

While reviewing the manuscript, I noticed in the section “3.5 Correlative analysis of environmental factors” that some data for the environmental factor had been incorrectly entered during the initial analysis and graphing. I have since input the correct data, conducted a new analysis and redrafted the figures, and included the updated results in the manuscript.

Similarly, due to some incorrect inputs of environmental factors initially, there were also issues with the analysis in the section "3.6 Random forest analysis". Therefore, after inputting the correct data, I conducted a new analysis and redrafted the figures, and have included the updated results in the manuscript.

I have made improvements to the discussion section, focusing on and emphasizing the key findings. I have eliminated redundant parts and conducted an in-depth analysis of how “variations in sediment particle size” influences the nitrogen cycle. The discussion and conclusion sections have been thoroughly revised to highlight the critical findings of this study.

Reviewer #2: The research work described the Characterization of Nitrogen-Cycling Microorganisms and Genes in Sediments. The article reports interesting results, however write up of the article could be a lot better than this draft. Language of introduction could be improved. There are so many sub-sections in the methodology section which do not follow the article style. A huge number of figures are also not desired. The quality of Figure 2a is not readable. Nitrogen is a very crucial element across sediments, discussion section needs to be extensively improved considering the nitrogen importance across sediments.

Reply:

I have improved the language in the introduction section. The methods section has been revised by removing unnecessary subsections to streamline the presentation and improve clarity. A significant amount of analysis and figures with weak relevance to this study have been removed. I have enhanced the quality of the figures and included all high-definition figures in the appendix. The discussion section has been thoroughly modified.

---

## [Editor Report · Decision Letter 1]

Dear Dr. Yang,

Thank you for submitting your manuscript to PLOS ONE. After careful consideration, we feel that it has merit but does not fully meet PLOS ONE’s publication criteria as it currently stands. Therefore, we invite you to submit a revised version of the manuscript that addresses the points raised during the review process.

We look forward to receiving your revised manuscript.

Kind regards,

Farhan Hafeez, Ph.D.

Academic Editor

PLOS ONE

Additional Editor Comments:

Dear Authors,

The revised version of the manuscript has been reviewed. Thanks to the authors for taking into account the previous comments. However, there are still the comments to be addressed as in appended text. The important comment: “Further, the figures could be merged, or only key data may be included instead of having huge numbers which could easily be merged” yet needs to be considered. Yes, the authors have just combined the figures, rather there is need to merge the figures for example in the form of panel. Currently, data presentation is more like a thesis document.

Sincerely,

---

## [Author Response · Author response to Decision Letter 2]

18 Feb 2025

Reviewer #1: The manuscript ‘Characterizing Nitrogen-Cycling Microorganisms and Genes in Sediments of the Three Gorges Reservoir’ focuses an important topic of investigating microbial-driven nitrogen cycle processes in the sediments of the Three Gorges Reservoir and assess the overall state of nitrogen cycling. The current manuscript comes in the domain of the journal and I appreciate the authors for their research work. However, due to various issues across different sections of manuscript, it needs revision. Overall, sentence structure needs to be improved. Some of general comments/questions are listed below while more specific questions and comments are inserted in the attached PDF file.

In abstract elaborate briefly, what authors want to say when write ‘46 types of nitrogen conversion functional genes’, nitrogen conversion not accurate there. Further. it’s not ‘refine’ rather this study aims to investigate or report. Please rephrase. The current structure of the introduction is not well organized. Additionally, the last part of the introduction needs to be revised, considering the main theme/objectives and findings of the study. Materials and Methods are very descriptive, it’s more like a thesis document especially in methods section, there are a large number of sub-headings in methods. And the representation of data also needs to be extensively improved. Further, the figures could be merged, or only key data may be included instead of having huge numbers which could easily be merged. The results are explained while the Discussion section could be improved, for example in addition to discussing the results in line with previous reports, key findings need to be discussed and highlighted. The section 4 overall, and specifically with focus on ‘ammonia oxidation requires specific environmental conditions’ is also like a thesis document, much detail and redundant rather than just discussing the key findings. Another very crucial aspect of the work i.e. how the ‘variations in sediment particle size’ influences the N cycling is described but lacks in depth analyses, for example, the relative abundance of these functional guilds across various environments could also be compared to enlighten the current results. Again, like other sections of the manuscript, Discussion, and Conclusions sections also needs to be comprehensive (instead of detailed paras and redundancy at places), the focus could only be on the key findings of the work.

Reply:

I have already made modifications to all the problematic areas listed in the PDF file according to the comments.

I have rewritten the abstract section, reorganizing the language. The sentence "46 types of nitrogen conversion functional genes" has been removed, and the structure of the sentences has been reorganized to ensure accurate expression.

I have revised the last part of the introduction, incorporating the research theme/objectives and the investigation results.

I have simplified the descriptions in the "Materials and Methods" section and consolidated some overly detailed sub-headings to reduce the complexity of the methods part and ensure that each section is directly relevant and contributes to the research findings.

I have eliminated certain analyses that bear relatively low relevance to the overall conclusion and opted to present solely the key data. By doing so, I aim to ensure the effective conveyance of the information's essence while circumventing excessive and redundant data. I have endeavored to render the data presentation more concise and influential, thus conforming to the standards expected of high-quality research papers.

While reviewing the manuscript, I noticed in the section “3.5 Correlative analysis of environmental factors” that some data for the environmental factor had been incorrectly entered during the initial analysis and graphing. I have since input the correct data, conducted a new analysis and redrafted the figures, and included the updated results in the manuscript.

Similarly, due to some incorrect inputs of environmental factors initially, there were also issues with the analysis in the section "3.6 Random forest analysis". Therefore, after inputting the correct data, I conducted a new analysis and redrafted the figures, and have included the updated results in the manuscript.

I have made improvements to the discussion section, focusing on and emphasizing the key findings. I have eliminated redundant parts and conducted an in-depth analysis of how “variations in sediment particle size” influences the nitrogen cycle. The discussion and conclusion sections have been thoroughly revised to highlight the critical findings of this study.

Reviewer #2: The research work described the Characterization of Nitrogen-Cycling Microorganisms and Genes in Sediments. The article reports interesting results, however write up of the article could be a lot better than this draft. Language of introduction could be improved. There are so many sub-sections in the methodology section which do not follow the article style. A huge number of figures are also not desired. The quality of Figure 2a is not readable. Nitrogen is a very crucial element across sediments, discussion section needs to be extensively improved considering the nitrogen importance across sediments.

Reply:

I have improved the language in the introduction section. The methods section has been revised by removing unnecessary subsections to streamline the presentation and improve clarity. A significant amount of analysis and figures with weak relevance to this study have been removed. I have enhanced the quality of the figures and included all high-definition figures in the appendix. The discussion section has been thoroughly modified.

---

## [Editor Report · Decision Letter 2]

Dear Dr. Yang,

Thank you for submitting your manuscript to PLOS ONE. After careful consideration, we feel that it has merit but does not fully meet PLOS ONE’s publication criteria as it currently stands. Therefore, we invite you to submit a revised version of the manuscript that addresses the points raised during the review process.

We look forward to receiving your revised manuscript.

Kind regards,

Farhan Hafeez, Ph.D.

Academic Editor

PLOS ONE

Journal Requirements:

Additional Editor Comments:

Dear Authors,

The revised version (R2) of the manuscript has been reviewed. I wonder, the authors just removed a few figures, while the comment was “Further, the figures could be merged, or only key data may be included instead of having huge numbers which could easily be merged”. For example, 3c, 3d, 5a and 6a are excluded. Either these were not important anymore or could these be merged (merged – means the data, the raw data could be combined to have composite figures). Currently, data presentation is more like a thesis document.

Sincerely,

---

## [Author Response · Author response to Decision Letter 3]

15 Apr 2025

Journal Requirements:

Reply:

I have checked the reference list to ensure it is complete and accurate.

Additional Editor Comments:

Dear Authors,

The revised version (R2) of the manuscript has been reviewed. I wonder, the authors just removed a few figures, while the comment was “Further, the figures could be merged, or only key data may be included instead of having huge numbers which could easily be merged”. For example, 3c, 3d, 5a and 6a are excluded. Either these were not important anymore or could these be merged (merged – means the data, the raw data could be combined to have composite figures). Currently, data presentation is more like a thesis document.

Reply:

Figures 3c, 3d, 5a, and 6a have been deleted because they are no longer important.

---

## [Editor Report · Decision Letter 3]

Characterizing nitrogen cycling microorganisms and genes in sediments of the Three Gorges Reservoir

PONE-D-24-37897R3

Dear Dr. Han,

We’re pleased to inform you that your manuscript has been judged scientifically suitable for publication and will be formally accepted for publication once it meets all outstanding technical requirements.

Kind regards,

Farhan Hafeez, Ph.D.

Academic Editor

PLOS ONE

Additional Editor Comments (optional):

Dear Author(s),

I write to you regarding the manuscript PONE-D-24-37897R3 entitled "Characterizing Nitrogen-Cycling Microorganisms and Genes in Sediments of the Three Gorges Reservoir" which was submitted to PLOS One.

Though I still wonder with the reply of the authors that ‘Figures 3c, 3d, 5a, and 6a have been deleted because they are no longer important’ in response to my comment ‘Either these were not important anymore or could these be merged’. However, most of the comments are addressed, hence the manuscript now carries the required potential. I am pleased to inform you that your manuscript is now sufficiently improved for possible publication in PLOS One. The formal acceptance is subject to fulfillment of all the technical requirements.

On behalf of PLOS One, I appreciate you for your contribution. Please keep us in mind for any future work that you consider to be appropriate for our readers.

Sincerely,

Farhan Hafeez, PhD
---

## [Editor Report · Acceptance letter]

PONE-D-24-37897R3

PLOS ONE

Dear Dr. Yang,

I'm pleased to inform you that your manuscript has been deemed suitable for publication in PLOS ONE. Congratulations! Your manuscript is now being handed over to our production team.

Kind regards,

on behalf of

Dr. Farhan Hafeez

Academic Editor

PLOS ONE